# Recurrence of Chronic Rhinosinusitis with Nasal Polyps After Surgery: Risk Factors, Predictive Models, and Treatment Approaches with a Focus on Western and Asian Differences

**DOI:** 10.3390/medicina61091620

**Published:** 2025-09-08

**Authors:** Yi-Shyue Chen, Chi-Yu Feng, Shih-Hao Su, Yu-Han Wang, Ting-Hua Yang, Chih-Feng Lin

**Affiliations:** 1Department of Otolaryngology, Head and Neck Surgery, National Taiwan University Hospital, Taipei 100225, Taiwan; yixuechen88@gmail.com (Y.-S.C.); thyang37@ntu.edu.tw (T.-H.Y.); 2Department of Medical Education, National Taiwan University Hospital, Taipei 100225, Taiwan; cyfeng0415@gmail.com (C.-Y.F.); music60807@gmail.com (S.-H.S.); yuhanwang0917@gmail.com (Y.-H.W.); 3Institute of Medical Device and Imaging, National Taiwan University, Taipei 100225, Taiwan

**Keywords:** chronic rhinosinusitis, nasal polyps, recurrence, endoscopic sinus surgery, eosinophilic inflammation, prediction models, biologics, regional differences

## Abstract

*Background and Objectives*: Chronic rhinosinusitis (CRS) frequently recurs following endoscopic sinus surgery (ESS), yet reported recurrence rates, risk factors, and treatment responses differ significantly across regions. This review aims to synthesize current evidence on recurrence patterns, predictive models, and treatment strategies, with a focus on comparing Asian and Western populations. *Materials and Methods*: A structured narrative review was conducted by searching PubMed, Embase, and Cochrane Library from January 2010 to June 2025. A total of 116 studies were included based on predefined criteria regarding recurrence definitions, risk factors, prediction models, and postoperative management. *Results*: Recurrence rates ranged from 12% to 76.6%, with wide variability attributed to differences in follow-up duration and recurrence definitions. Key risk factors included tissue eosinophilia, comorbid asthma, and type 2 inflammation. Asian predictive models emphasized inflammatory biomarkers such as tissue and blood eosinophils, whereas Western models incorporated imaging, prior surgical history, and symptom burden. While biologics are widely used in the West, their adoption remains limited in Asia, where endotype-driven corticosteroid strategies are predominant. *Conclusions*: CRS recurrence after ESS is influenced by inflammatory endotypes, comorbidities, and regional treatment paradigms. Cross-regional differences in immune profiles and healthcare access necessitate the development of standardized definitions and validated, endotype-driven prediction tools. Tailored treatment strategies, especially for non-type 2 CRS, are essential to achieving equitable and effective care globally.

## 1. Introduction

Chronic rhinosinusitis (CRS) is a prevalent and debilitating inflammatory disease of the paranasal sinuses, characterized by persistent sinonasal symptoms lasting for at least 12 weeks. The diagnostic criteria, which are generally consistent across regions and based on guidelines such as the European Position Paper on Rhinosinusitis and Nasal Polyps (EPOS) and the International Consensus Statement on Allergy and Rhinology: Rhinosinusitis (ICAR), require at least two of the following symptoms: nasal blockage/obstruction/congestion or discharge (anterior/posterior nasal drip), facial pain/pressure, or reduction/loss of smell. These must be supported by objective evidence of inflammation, such as nasal polyps, mucopurulent discharge, or edema on endoscopy, or mucosal changes on computed tomography (CT) within the ostiomeatal complex and/or sinuses. Endoscopic sinus surgery (ESS) is commonly performed in patients with CRS who fail to respond adequately to maximal medical therapy, aiming to improve sinus ventilation, restore mucociliary clearance, and alleviate symptoms. However, recurrence of CRS following ESS remains a major clinical challenge, with reported recurrence rates varying widely across studies.

Understanding the factors contributing to postoperative recurrence is crucial for optimizing surgical outcomes and guiding postoperative management. Numerous risk factors for recurrence have been identified, including tissue eosinophilia, nasal polyps, asthma, and allergic rhinitis [1,2,3]. Nevertheless, considerable heterogeneity exists among studies, particularly when comparing data from different geographic regions.

One of the most notable regional disparities lies between Asian and Western populations. Asian cohorts have been shown to exhibit a higher proportion of non-eosinophilic CRS and distinct inflammatory profiles compared to their Western counterparts [4,5]. These differences may influence not only recurrence rates but also the predictive value of established risk factors and scoring systems. Furthermore, recent advances in the treatment of recurrent CRS, including the use of biologic therapies targeting type 2 inflammation, have gained traction predominantly in Western countries, whereas access and adoption remain limited in many parts of Asia [5,6].

Given these disparities, a comprehensive synthesis of the current evidence is warranted to clarify the recurrence patterns of CRS after ESS, elucidate key risk factors and predictive models, and explore regional variations in treatment strategies. This review aims to provide an up-to-date overview of these domains, with particular emphasis on differences between Asian and Western populations.

## 2. Materials and Methods

### 2.1. Literature Search Strategy

A comprehensive literature search was conducted in PubMed, Embase, and the Cochrane Library to identify studies on CRS recurrence after ESS, focusing on recurrence rates, associated risk factors, predictive models, and postoperative management strategies. The search covered studies published from January 2010 to June 2025, with earlier seminal papers included when relevant for historical context. The full search strategy using MeSH terms and keywords was detailed in the Appendix A. In addition, reference lists of the included articles were manually screened to identify additional eligible studies.

### 2.2. Eligibility Criteria

Inclusion criteria:Studies involving adult patients (aged ≥ 18 years) undergoing endoscopic sinus surgery (ESS) for chronic rhinosinusitis (CRS)Reporting recurrence rates, recurrence risk factors, prediction models, or postoperative treatment strategiesConducted in or reporting data from Asian or Western populationsPublished in English with full text available

Exclusion criteria:Pediatric studies (involving patients < 18 years)Case reports or case series with <30 patientsNarrative reviews without systematic methodology, unless cited for backgroundConference abstracts, editorials, and letters

### 2.3. Study Screening and Selection

After removing duplicates, a total of 630 articles were initially selected for title and abstract screening. Of these, 163 articles were retained and subsequently subjected to full-text review for structured evaluation. Although this review employed a narrative synthesis approach, a structured selection strategy was implemented to ensure the inclusion of methodologically sound and clinically relevant studies. During full-text screening, preference was given to cohort studies with clear definitions of recurrence, adequate sample sizes, and detailed reporting of surgical or inflammatory characteristics. Finally, a total of 116 representative studies were included for detailed synthesis. These were chosen to reflect a balanced and clinically relevant perspective on recurrence of CRS after surgery, while allowing meaningful comparison between Asian and Western populations. Studies were not pooled for meta-analysis due to heterogeneity in definitions and outcomes, but conflicting findings were discussed narratively to highlight regional variations. Figure 1 illustrates study screening and selection.

## 3. Recurrence Rates of CRS After Surgery

### 3.1. Definitions of Recurrence

The definition of recurrence in CRS after ESS varied widely across studies, contributing to heterogeneity in reported recurrence rates and prognostic interpretations. Broadly, recurrence has been defined based on either subjective symptom, objective findings, or a combination of both.

Symptom-based definitions typically rely on the reappearance or worsening of nasal symptoms such as obstruction, discharge, or hyposmia, often measured using tools like the Sino-Nasal Outcome Test (SNOT-22) or visual analog scale (VAS) [1,2,4]. In contrast, objective definitions include endoscopic detection of polyp regrowth or radiological evidence of mucosal disease on computed tomography (CT), commonly scored via the Lund-Kennedy or Lund-Mackay systems [1,3,7]. Some studies, particularly from Asian cohorts, also incorporate histopathological criteria, such as elevated tissue eosinophil counts, to stratify recurrence risk [4,8,9,10].

### 3.2. Recurrence Rates in Western vs. Asia

In Asian cohorts, recurrence rates ranged from 13.6% to 65.3%, with 8 out of 12 studies reporting between 25% and 45%. Of these, 10 out of 12 were retrospective with short- to intermediate-term follow-up (typically 6–36 months). However, two recent studies have expanded this landscape. A multicenter retrospective study which followed up for 5 years across five Asian countries reported recurrence rates of 63.3% in Korea and 33.3% in Japan, underscoring the variability within Asia [11]. Meanwhile, a prospective cohort study with an 8-year follow-up in China showed an overall recurrence rate of 21.8% but noted a significantly higher rate (72.7%) among patients with type 2 inflammatory endotypes [12].

By contrast, Western studies reported recurrence rates ranging from 12.0% to 76.6%, including several long-term cohort studies. For instance, Vlaminck et al. [13] reported a 59.1% recurrence rate at over 10 years, and Calus et al. documented 76.6% after 12 years [14]. Even shorter-term studies frequently exceeded 30% recurrence, particularly in patients with eosinophilic CRSwNP, asthma, or prior revision surgery [15,16].

The observed regional disparities in recurrence rates are likely attributable to methodological heterogeneity, including variations in follow-up duration and criteria used to define recurrence, rather than intrinsic biological differences. While recent Asian studies have mitigated prior limitations by incorporating extended observation periods, the use of differing definitions—objective markers such as revision surgery and endoscopic scores in Western studies versus symptom burden and inflammatory biomarkers in Asian cohorts—continues to hinder direct cross-regional comparisons. Furthermore, disparities in access to biologic therapies, such as anti-IL-5 or anti-IgE agents, may affect recurrence prevention strategies and their integration into surgical planning. Figure 2 demonstrates the recurrence rates of CRSwNP in Western and Asian cohorts across representative studies; however, disparities in recurrence rates are primarily due to methodological heterogeneity—different follow-up durations, definitions of recurrence, and use of objective versus subjective recurrence criteria. Meanwhile, regional access to advanced biologic therapies may also contribute to differences in surgical outcomes and recurrence prevention strategies.

## 4. Risk Factors of CRS Recurrence

The risk factors can be broadly categorized into patient-related characteristics, disease-specific traits, and surgical variables. While several risk factors appear consistent across regions, others exhibit notable geographical variation in prevalence, diagnostic emphasis, or predictive value.

### 4.1. Patient Factors

Among patient-related characteristics, asthma and aspirin-exacerbated respiratory disease (AERD) have been consistently associated with increased risk of CRS recurrence [3,17]. Both Asian and Western studies have reported significantly higher recurrence rates among asthmatic patients, with odds ratios ranging from 1.6 to 2.9 [4,18,19]. In Taiwan, a population-based study showed that patients with both asthma and CRSwNP had a 1.7-fold increased risk of revision ESS over an 18-year period [20]. Similarly, AERD has been linked to aggressive, treatment-refractory forms of CRSwNP, with recurrence rates exceeding 50% in several cohorts [3,21].

Allergic rhinitis (AR) is another commonly cited risk factor, often accompanying asthma or nasal hyperreactivity. Its contribution to recurrence appears to be mediated through increased mucosal edema, higher eosinophilic infiltration, and suboptimal symptom control [18,22,23]. Asian studies in particular have highlighted the synergistic effect of AR and asthma on recurrence risk, reporting odds ratios as high as 2.5 in a subpopulations [24].

Several studies have highlighted smoking as a negative prognostic factor [25,26,27,28]. A large U.S. database study involving over 34,000 CRS patients found that tobacco use independently increased the risk of revision ESS, especially among those with comorbid asthma (adjusted OR 1.72) [29], and a Taiwanese analysis found that smoking predicted persistent postoperative inflammation despite biologic therapy [25].

Environmental exposures have also been implicated in disease recurrence. Occupational exposure to inhaled dusts—both organic (e.g., cotton, wood) and inorganic (e.g., bleach, cement, metal fumes)—has been identified as an independent risk factor for postoperative recurrence in patients with CRSwNP. A prospective study found that dust-exposed workers had a significantly higher recurrence rate after ESS (*p* = 0.001), even after adjusting for age, sex, asthma subtype, allergic rhinitis, smoking, and radiologic severity [30]. In a large case–control study, the likelihood of occupational exposure increased with the number of FESS procedures, and patients exposed to low molecular weight irritants were significantly more likely to require revision surgeries (adjusted OR = 1.64–1.97) [31]. These findings suggest that long-term environmental exposure may contribute to persistent inflammation and surgical failure in susceptible individuals.

Other demographic and clinical factors show more variable associations. For instance, Influence of sex and age remains inconsistent across studies. Western data suggest a slightly higher recurrence risk in females, possibly reflecting hormonal or healthcare access differences [32,33], while other studies report no significant sex-based differences [34]. Age-related patterns appear region-dependent; younger patients were associated with higher recurrence in some Asian studies [20], whereas others identified older age as a risk factor, potentially due to immunosenescence and delayed mucosal healing [35]. A recent study stratifying patients by both age and sex found the highest recurrence rate in young-adult males, suggesting a possible interaction between demographic and underlying inflammatory pattern [36]. In Western cohorts, Cystic Fibrosis (CF) has also emerged as a particularly strong predictor of revision. A U.S. population-based study involving over 34,000 CRS patients found that those with CF had significantly higher revision rates (18.7% vs. 13.4%, *p* < 0.001), and CF remained an independent predictor even after adjusting for asthma, nasal polyps, allergy, and smoking status (aOR = 2.18) [26].

In recent years, metabolic comorbidities such as obesity, metabolic syndrome, and type 2 diabetes mellitus (T2DM) have emerged as modifiable risk factors, especially in Chinese cohort. Xie et al. demonstrated that each unit increase in body mass index (BMI) was associated with a 14.3% increase in recurrence odds [28]. Another study concluded that overweight and obesity aggravated tissue eosinophil infiltration, and IL-5 and IL-17A expressions contributing to the recurrent mechanisms of CRSwNP [37]. One prospective study further showed that overweight and obese patients exhibited higher symptom burden and a mixed type 2/type 3 inflammatory profile, with BMI emerging as the only consistent predictor of recurrence across weight categories [38]. In parallel, metabolic syndrome has also been shown to independently increase postoperative recurrence risk, accompanied by elevated eosinophil counts and enhanced IL-5 and IL-17A expression, with recurrence risk rising proportionally with the number of metabolic syndrome components [39]. Moreover, metabolic syndrome and T2DM have been associated with impaired wound healing and persistent inflammation, leading to higher recurrence rates compared to non-affected individuals [40,41]. Given the metabolic relevance of serum uric acid, accumulating evidence suggests it may serve as a novel prognostic biomarker, with both prospective and retrospective Chinese cohorts reporting significantly higher recurrence risk in patients with hyperuricemia [22,42].

### 4.2. Disease Characteristics

Disease-specific traits, particularly inflammatory endotypes and histopathological markers, are central risk factors. The distinction between CRSwNP and CRSsNP is fundamental, with numerous studies confirming that CRSwNP is associated with a higher risk of postoperative recurrence [20,21,32,43]. A pan-European study reported a threefold increase in revision surgery rates among patients with CRSwNP compared to those with CRSsNP [32].

Eosinophilic inflammation represents one of the most consistent predictors of poor outcomes. High tissue eosinophil counts have been associated with increased recurrence rates in both Asian and Western populations [4,44]. Region-specific tools, such as the Japanese Epidemiological Survey of Refractory Eosinophilic Chronic Rhinosinusitis (JESREC) scoring system [45] developed in Japan, have been widely applied in Asia to stratify patients based on eosinophilic burden, comorbid asthma, and radiographic findings [9]. However, the performance of JESREC in Western populations may be limited due to the near-universal presence of eosinophilia in CRS patients in those regions [1].

Inflammatory endotyping has further revealed regional differences in immune profiles. Biomarkers such as blood eosinophil counts, serum immunoglobulin E (IgE), and eosinophil cationic protein (ECP) have demonstrated prognostic value in both regions [24,35,46,47,48,49], although the thresholds used for clinical interpretation may vary. Type 2 inflammation—characterized by eosinophilic dominance and cytokines such as IL-5 and IL-13—is more commonly observed in Western cohorts and is associated with severe disease and high revision rates [47]. Notably, a long-term Italian cohort found that both neutrophilic and eosinophilic nasal cytology were associated with markedly elevated 10-year recurrence rates (88% and 100%, respectively), compared to only 59% in patients with normal cytology, further supporting the prognostic relevance of inflammatory cell type [50].

In contrast, several Asian studies have identified a higher prevalence of non-type 2 or mixed eosinophilic-neutrophilic inflammation [8,51], highlighting the endotypic heterogeneity of this population and the associated variability in treatment response and long-term prognosis. In a Chinese cohort with recurrent CRSwNP and asthma followed for 8 years, cluster analysis based on inflammatory cytokine profiles revealed that type 2 inflammation was associated with the highest recurrence, while a distinct non-type 2 cluster showed elevated ECP/MPO ratios was still experienced high recurrence [11]. These findings may have implications for the generalizability of biomarker-based models and for therapeutic decision-making.

### 4.3. Surgical Factors

Surgical technique and postoperative management are critical determinants of long-term ESS success. Incomplete surgery—especially inadequate opening of the ethmoid, frontal, or sphenoid sinuses—has been consistently identified as a leading cause of recurrence in both Asian and Western studies [32,52]. Accordingly, studies have reported that more extensive procedures—such as complete ethmoidectomy or full-house sinus surgery—are associated with improved prognosis compared to limited interventions [21,33].

Detailed anatomical analyses in revision cases have revealed high frequencies of residual obstructive structures. For example, a German cohort of 253 CRS patients undergoing revision ESS found that incomplete anterior ethmoidectomy (51%), residual uncinate process (37%), and recirculation phenomenon (33%) were among the most common contributors to disease persistence [53]. Another study also showed that undissected or inflamed retromaxillary cells, which are often overlooked during ethmoid dissection, have also been identified as anatomical risk factors for recurrence [54]. These findings underscore technical limitations, rather than disease biology alone, often drive surgical failure.

The extent of the initial procedure and the surgeon’s experience are also key factors influencing outcomes. Both Asian and Western study revealed significantly lower revision rates among patients operated on by high-volume ESS surgeons, suggesting that surgical experience plays a critical role in long-term outcomes [40,55].

On the other hand, effective control of postoperative inflammation is essential for preventing recurrence. Across regions, early mucosal healing, minimal crusting, and optimal use of topical corticosteroids have been linked to improved outcomes [7,22,47]. Inadequate suppression of eosinophilic inflammation during the early healing phase may allow residual inflammatory foci to re-expand, thereby increasing the risk of recurrence [48]. Serial postoperative endoscopy provides an objective tool for monitoring disease control [56]. Scoring systems such as the modified Lund-Kennedy and Perioperative Sinonasal Endoscopic (POSE) scores have been employed in both Asian and Western cohorts to identify early signs of relapse [7,57]. Elevated endoscopic scores within 3 to 6 months after surgery have been associated with subsequent recurrence, reinforcing their value as predictive instruments [7,9,57]. Table 1 summarizes the risk factors of CRS recurrence.

## 5. Predictive Models for Recurrence

### 5.1. Overview of Predictive Modeling Approaches

A wide spectrum of predictive factors has been identified for recurrence of CRS after ESS, and various models have been proposed based on clinical, serologic, histologic, radiologic, and integrative parameters. These models often reflect regional priorities and available diagnostic tools.

### 5.2. Clinical Predictors and Symptom-Based Models

Building on the aforementioned risk factors, various patient- and disease-related features have been identified as clinical predictors of CRS recurrence and revision surgery. Comorbid asthma, allergic rhinitis, NSAID-exacerbated respiratory disease (NERD), and previous surgery have been consistently associated with increased risk of postoperative recurrence and the need for revision ESS [30,58,59,60,61,62,63]. Clinical symptoms may also provide prognostic value. One study highlighted the SNOT-22 trajectory as a dynamic predictor of revision risk, showing that lack of improvement by 3 months or worsening between 3 and 12 months postoperatively was significantly associated with subsequent revision surgery [64].

The time interval between previous surgeries has also been linked to prognosis, with shorter intervals predicting higher recurrence rates [65]. Some cohorts have developed clinical composite models to predict recurrence, integrating symptoms, compliance, and clinical endotype [7,66,67]. These models lay the groundwork for more sophisticated prediction systems.

### 5.3. Serologic and Immunologic Biomarkers

Serologic biomarkers are frequently explored, especially in Asian cohorts. Peripheral eosinophil count, basophil count, and their respective ratios—e.g., eosinophil-to-lymphocyte ratio (ELR), neutrophil-to-lymphocyte ratio (NLR), serum levels of eosinophil cationic protein (ECP)—have been associated with postoperative recurrence [48,68,69,70,71,72,73], although no standardized or validated cutoff values have been established. Beyond traditional serologic markers, recent studies have highlighted the role of immune mediators in predicting recurrence. Elevated levels of cytokines involved in immune regulation and tissue remodeling—such as interleukins, eotaxin, complement proteins —have been identified in recurrent CRS, with moderate predictive accuracy [74,75,76]. In parallel, immune cell subsets including regulatory T cells and innate lymphoid cells have shown promise in refining endotype classification and informing risk stratification [77,78].

### 5.4. Histologic Markers of Inflammation

Histologic parameters remain central to many Asian studies, particularly tissue eosinophilia, which consistently predicts recurrence. Several studies have proposed various thresholds, including absolute counts such as >55 eosinophils per high power field (HPF) or proportion-based cutoffs like ≥27% of total inflammatory cells, as strong predictors for postoperative recurrence [9,79]. Some investigators favor percentage-based assessments due to variability in tissue architecture [79]. In addition, markers of eosinophilic activation such as Charcot-Leyden crystals (CLC)—quantified via tissue histology or CLC mRNA levels in nasal brushing—had demonstrated promising predictive value [80,81]. Histological signatures such as IL-5 and IL-13 expression, dense eosinophilic infiltration, and eosinophilic mucin have also been associated with recurrence risk and unfavorable surgical outcomes [59,70,79,82,83]. In addition to eosinophilic indicators, recent studies have highlighted the role of mast cell burden. A U.S. prospective study reported that higher epithelial and stromal mast cell densities were significantly associated with earlier recurrence [84].

### 5.5. Radiologic Predictors

Radiologic scores, particularly the Lund-Mackay CT score and ethmoid-to-maxillary (E/M) opacification ratio, have been associated with disease severity and recurrence [85,86,87,88,89]. A higher ethmoid dominance or total score typically portends poorer prognosis. Central Compartment Atopic Disease (CCAD)—a subtype defined by central ethmoid-predominant opacification—has been associated with significantly lower polyp recurrence (7.9%) and revision rates (5.3%) compared to other CRSwNP variants such as AFRS or AERD, despite its ethmoid-heavy radiologic presentation [90]. This underscores the importance of interpreting radiologic severity in the context of underlying immunologic endotype. The E/M opacification ratio is a more consistent predictor for recurrence in Asian/non-type 2 CRS and a marker of olfactory recovery in Western/type 2 CRS populations. Radiological severity predicts recurrence in both regions but with different modifying factors.

### 5.6. Role of Microbiota in Recurrence Risk

A few studies have suggested that microbiota composition may also influence recurrence. For instance, rectal *Staphylococcus aureus* carriage and specific nasal microbial profiles were associated with higher risk of postoperative relapse, underscoring a possible role of host–microbe interactions in CRS pathophysiology [91,92]. In addition, the presence of *S. aureus* on sinus culture—either preoperatively or four months after surgery—was independently associated with higher recurrence risk in a prospective cohort of high-risk CRS patients, particularly those undergoing revision ESS [93].These findings support the hypothesis that *S. aureus* may interfere with mucosal healing and immune responses, and serve as a potential biomarker for disease persistence. *S. aureus* and microbiota data suggest region-specific patterns influencing recurrence.

### 5.7. Integrated and Machine Learning–Based Prediction Models

Integrated prediction models combining multiple domains have been increasingly proposed in both Western and Asian cohorts. These models typically incorporate combinations of symptom burden (e.g., VAS, SNOT-22), radiologic severity (e.g., Lund-Mackay score), comorbidities (e.g., asthma, allergic rhinitis), and inflammatory biomarkers such as peripheral or tissue eosinophils. In Asia, several models based on logistic regression have demonstrated strong predictive value. For example, one model incorporated allergic rhinitis, olfactory impairment VAS, facial pain VAS, Lund-Mackay score, and eosinophil percentage [18], while another constructed a nomogram integrating age, IL-6, IL-8, and blood eosinophils [94]. Western studies similarly developed multivariable models using factors such as age < 55, prior ESS, AERD, eosinophils ≥ 300/μL, and LMS > 17 to stratify revision risk [95]. A systematic review further concluded that no single biomarker suffices; rather, models integrating clinical, radiologic, and inflammatory data yield superior accuracy [8]. In parallel, machine learning–based approaches are emerging, incorporating multidimensional data such as miRNAs, cytokines, and medication history to develop predictive tools with accuracies exceeding 80% [7]. Table 2 illustrates the predictive model for recurrence.

## 6. Management of Recurrent CRS

### 6.1. Medical Treatments

Medical therapy remains a cornerstone in the management of recurrent CRS. Corticosteroids, both systemic and topical, are widely used to control symptoms and reduce polyp burden. However, their long-term use raises concerns about systemic side effects, prompting a shift toward more targeted therapies.

In Western countries, biologic therapies have become a key strategy for recurrence control in recent years. Dupilumab, targeting the IL-4 and IL-13 pathways, has demonstrated consistent benefits in reducing nasal polyp scores, improving olfaction, and delaying the need for revision surgery [47,96,97,98,99]. Furthermore, systematic reviews indicate that patients’ treatment preferences increasingly favor efficacy-driven approaches, with over half of respondents prioritizing biologics or newer implants when cost is not a barrier [100].

In Asian populations, medical therapy tends to follow a more conservative stepwise approach. Biologics are used more selectively, typically reserved for refractory patients with confirmed type 2 inflammation, due to both cost and regulatory constraints [5,6]. Nevertheless, growing interest in their use is evident. A Taiwanese cohort study including only patients with tissue eosinophilia demonstrated that postoperative dupilumab led to significantly greater improvements in endoscopic scores and symptom burden [25]. In addition, a subgroup analysis of Japanese participants in the SINUS-52 trial revealed that dupilumab treatment resulted in significant improvements in nasal polyp score, nasal congestion, and Lund-Mackay CT scores at week 24 compared to placebo, with benefits sustained over 52 weeks. The safety profile in the Japanese cohort was consistent with the global population, with nasopharyngitis being the most common adverse event [101].

However, these findings apply primarily to patients with type 2 inflammation. Asian CRS patients more commonly exhibit non-type 2 endotypes, including neutrophilic, type 1, and type 3 inflammation, which are less responsive to T2-targeted biologics. A recent review to Asian patient emphasized that nearly half of Asian cases are non-eosinophilic, underscoring the need for careful patient selection and alternative treatments beyond the T2 pathway [5]. Moreover, the recent study from Japan have also identified corticosteroid-responsive subgroups, among patients with shorter disease duration prior to initial surgery [102]. These findings support the value of individualized, endotype-driven treatment plans in different regional contexts.

### 6.2. Surgical Revision Strategies

Surgical revision is indicated for patients with persistent or recurrent disease despite optimal medical therapy. The timing and extent of revision vary, typically based on symptom severity, imaging findings, and endoscopic evidence of polyp regrowth. Endoscopic sinus surgery (ESS) encompasses a range of techniques for treating chronic rhinosinusitis. Radical ESS is characterized by extensive mucosal and bone removal, aiming to eradicate disease and minimize recurrence, and is generally reserved for refractory or severe cases. The Draf 3 procedure, typically performed in conjunction with ESS, involves a modified Lothrop frontal sinusotomy to achieve maximal frontal sinus drainage, especially for complex or recurrent frontal sinus pathology. Conventional functional endoscopic sinus surgery (FESS) and traditional ESS are less extensive, focusing on restoring natural sinus ventilation by selective removal of diseased tissue, which results in faster recovery but higher early recurrence. Reboot surgery is a more radical, non-mucosa-sparing technique involving complete mucosal removal down to the periosteum, targeting refractory chronic rhinosinusitis with nasal polyps (CRSwNP) unresponsive to previous therapies, significantly lowering polyp recurrence and improving long-term outcomes.

Surgical principles include clearing residual ethmoid cells, enlarging frontal sinusotomies, and optimizing drainage pathways [103]. Supporting this principle, a prospective Chinese cohort of patients with recurrent CRSwNP and asthma demonstrated that those who underwent more extensive procedures—radical ESS or ESS combined with Draf 3—had significantly lower revision rates at 5 years (17.4% and 16.0%, respectively) compared to those treated with conventional FESS (45.2%) [11]. Similarly, partial “reboot” ESS, a non-mucosa-sparing approach, was shown in an Italian cohort to significantly reduce recurrence rates compared to traditional ESS (e.g., 24-month recurrence: 27.3% vs. 90.9%) and improved symptom scores while reducing systemic corticosteroid reliance [104]. These findings underscore the importance of comprehensive initial surgical intervention, especially in patients with severe or eosinophilic disease.

In Western countries, reported revision rates after ESS range from 20% to 30% in long-term studies, which was higher than Asian populations. Guidelines such as EPOS 2020 recommend earlier surgical intervention in patients with severe or recurrent type 2 inflammation, aiming to improve outcomes and reduce relapse risk. Multimodal strategies—incorporating revision ESS, long-term topical corticosteroids, and biologics—are routinely employed, supported by both clinical evidence and favorable reimbursement structures [47].

In contrast, a more conservative, stepwise treatment paradigm is often adopted in Asian settings, where revision surgery is less frequently pursued. A nationwide Taiwanese cohort reported a 14.5% revision rate with a mean interval of 5.9 years post-surgery and identified younger age and comorbid asthma as key predictors [20]. Cultural values and healthcare system limitations in Asia may further contribute to the underutilization of revision surgery. Notably, a U.S.-based retrospective study found that among racially diverse patients undergoing ESS, Asian individuals had the lowest 2-year revision rate (6.3%), despite comparable symptom improvement across all racial groups [105]. Similarly, another study focusing on Asian American patients with medically refractory CRS reported that they were significantly less likely to undergo ESS compared to non-Asian Americans (40.0% vs. 81.0%, *p* = 0.005), even after adjusting for symptom severity and disease burden [106].

Beyond surgical technique, adjuvant interventions aimed at modulating postoperative inflammation have also gained increasing attention. Among these, steroid-eluting stents (SES) have emerged as a valuable tool for enhancing mucosal healing and reducing polyp regrowth in the early postoperative period [107,108]. Clinical trials and real-world evidence support their use in reducing adhesions, facilitating ostial patency, and ultimately lowering the likelihood of revision surgery in high-risk patients [109,110,111,112]. Real-world data suggest that steroid-eluting implants offer comparable short-term healthcare utilization and may provide favorable cost-effectiveness compared to revision surgery in U.S. settings [113]. Complementing these findings, a randomized, controlled study demonstrated that in-office placement of steroid-eluting implants yielded sustained symptomatic and endoscopic improvements over 6 months, significantly reducing the need for further surgical intervention in patients with recurrent nasal polyposis [114]. Moreover, treatment trajectories appear to diverge: implant users tend to transition toward biologic therapies, while surgical patients are more likely to require further revision ESS, highlighting differences in long-term care patterns [115].

These disparities highlight the need to reassess surgical candidacy criteria and to advance alternative anti-inflammatory approaches tailored to populations underrepresented in existing treatment frameworks.

## 7. Discussion

### 7.1. Methodological Limitations

This scoping review highlights the heterogeneous and evolving nature of current research on CRS recurrence following ESS. A key methodological challenge is the absence of standardized criteria for defining recurrence and assessing outcomes. Studies vary widely in their use of symptom-based assessments, endoscopic scores, radiologic staging, and, less frequently, histopathological or immunological markers. This inconsistency impedes direct comparisons and limits the development and generalizability of predictive tools.

A recent systematic review of long-term ESS outcomes similarly emphasized the need for unified definitions and standardized outcome reporting. The authors advocated for consistent use of tools such as SNOT-22, nasal polyp scores, validated olfactory tests, and clearly defined surgical parameters to enable meaningful cross-study comparisons and facilitate integration of real-world and clinical trial data [116].

In addition, much of the existing evidence is derived from retrospective, single-center studies. Differences in study design, cohort composition, and outcome measurement further contribute to the fragmented nature of the literature.

### 7.2. Regional Differences in Predictors and Models

To further illustrate the key differences between Asian and Western populations in the context of CRS recurrence, a summary comparison table is provided below (Table 3). This table synthesizes the dominant inflammatory patterns, clinical implications, treatment paradigms, and data support across regions, offering a concise reference for cross-regional evaluation.

Notably, most available prediction models have been developed in Western populations and center around type 2 inflammation, particularly eosinophilic CRS. While informative, this focus may not fully capture the spectrum of CRS endotypes encountered globally. In Asian cohorts, for example, non-type 2 or mixed eosinophilic–neutrophilic inflammation is more frequently observed, which may influence treatment response and risk stratification. Rather than representing a limitation, these differences reflect the broader immunopathological diversity of CRS and highlight the need for region-specific or endotype-adapted predictive frameworks.

### 7.3. Therapeutic Implications

Therapeutic approaches to CRS vary considerably across regions. In Western countries, biologic therapies are widely adopted—often supported by insurance coverage—and are frequently integrated early alongside revision surgery and topical steroids. In contrast, Asian countries tend to follow a more conservative, stepwise strategy. Biologics are more selectively prescribed, usually for patients with clearly defined type 2 inflammation, and revision surgery is often deferred based on clinical judgment, patient preference, and system-level constraints. These differences reflect not only disparities in healthcare resources but also culturally informed clinical decision-making.

A growing recognition of non-type 2 CRS, particularly in Asia, underscores a persistent therapeutic gap. These patients, often presenting with neutrophilic or mixed inflammatory patterns, respond poorly to type 2–targeted biologics and remain underserved due to a lack of validated predictive models and tailored treatment options. Emerging strategies such as anti–IL-17, anti-TSLP therapies, microbiome modulation, and transcriptomic profiling may help expand therapeutic possibilities and support a more inclusive management paradigm. Meanwhile, in Western settings, the widespread use of biologics raises important concerns about long-term cost-effectiveness, optimal treatment duration, and overtreatment risks in borderline cases. Although clinical trial data are encouraging, real-world evidence on sustained disease control and biomarker-guided tapering remains limited.

Moving forward, shared decision-making frameworks that integrate patient preferences, disease severity, inflammatory endotype, and real-world outcomes are essential. Recent guidelines such as EPOS 2020 and EUFOREA have emphasized the need for individualized, stratified treatment approaches [6]. To ensure equitable and sustainable CRS care, future efforts should focus on cross-regional validation of prediction tools and development of novel therapeutic pathways for non-type 2 CRS.

## 8. Conclusions

The recurrence of CRS after ESS remains a multifactorial challenge influenced by inflammatory endotypes, patient comorbidities, surgical techniques, and regional treatment paradigms. While eosinophilic inflammation and comorbid asthma are consistently recognized predictors across studies, the prognostic landscape is further complicated by geographic variations in immune profiles, clinical practices, and access to biologics. Asian cohorts tend to emphasize histological and serological biomarkers, while Western approaches favor integrated models combining radiologic, clinical, and patient-reported outcomes. These differences underscore the need for standardized recurrence definitions and globally validated, endotype-specific prediction tools. To advance equitable care, treatment strategies must evolve to address non-type 2 CRS and ensure access to tailored interventions across regions. Future research should prioritize cross-cultural validation, inclusion of diverse inflammatory endotypes, and the integration of real-world data to optimize long-term outcomes in CRS management.

## Figures and Tables

**Figure 1 medicina-61-01620-f001:**
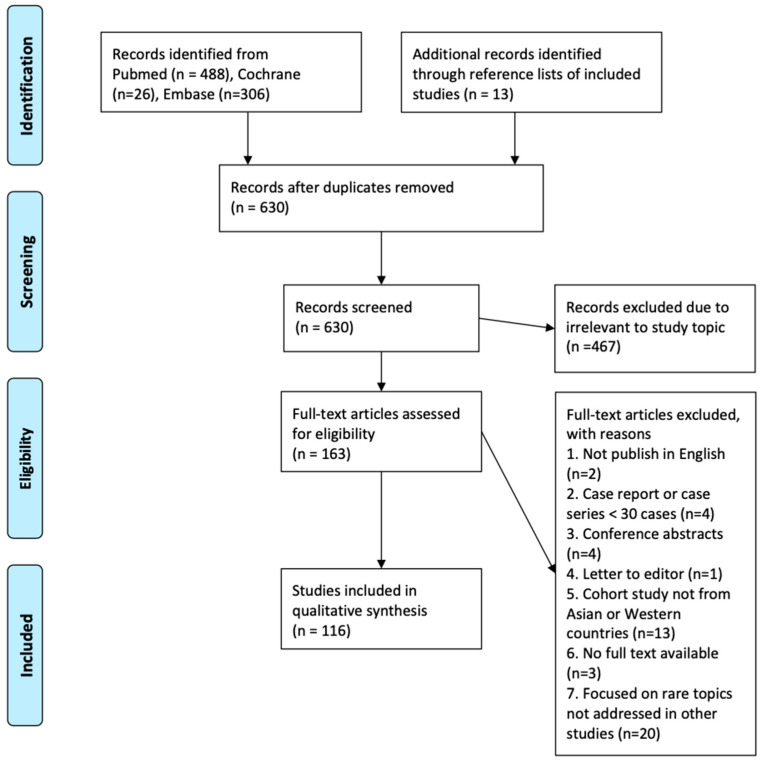
Study screening and selection.

**Figure 2 medicina-61-01620-f002:**
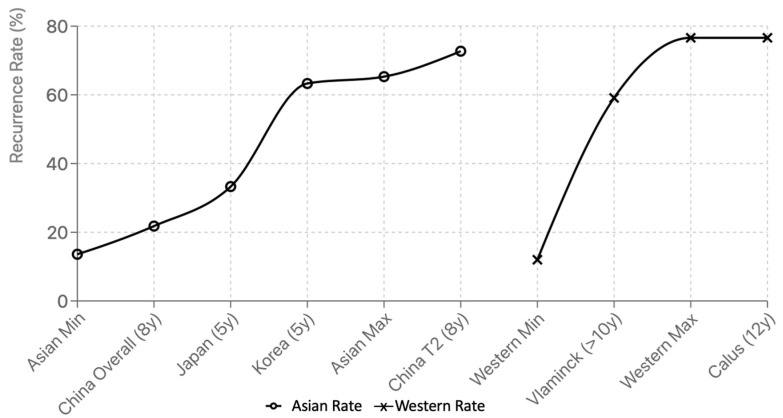
Recurrence rates after CRSwNP surgery: Western vs. Asian cohorts.

**Table 1 medicina-61-01620-t001:** Risk factors of CRS recurrence.

	Risk Factor/Description
Patient Factors	Asthma and AERD are common and strong predictors for CRS recurrence [3,4,18,19,20,21]
	Allergic rhinitis (AR), often combined with asthma, increases recurrence risk via mucosal edema and eosinophilia [18,22,23,24]
	Smoking increases risk, especially among asthmatic patients; also predicts postoperative inflammation [25,26,27,28,29]
	Environmental and occupational exposures (e.g., dust, chemicals) independently raise recurrence and revision surgery risk [31]
	Demographics: female sex and age (either younger or older, region-dependent) may influence recurrence risk [32,33,34,35,36]
	Cystic fibrosis (CF) is a strong predictor of revision, especially in Western cohorts [26]
	Metabolic comorbidities (obesity, metabolic syndrome, T2DM): higher BMI, MetS components, and T2DM linked to recurrence [28,37,38,39,40,41]
	Serum uric acid (hyperuricemia) identified as an independent risk biomarker for recurrence [22,42]
Disease Characteristics	CRSwNP carries a higher recurrence risk than CRSsNP; pan-European study shows 3× higher revision with polyps [20,21,32,43]
	Eosinophilic inflammation (tissue/blood eosinophilia, high ECP, high IgE, high IL-5) is a consistent predictor of relapse [1,4,44,45]
	Inflammatory endotype (type 2 inflammation, high IL-5/IL-13) is associated with more severe, recurrent disease [24,35,46,47,48,49,50]
	Non-type 2 or mixed inflammation (eosinophilic-neutrophilic) is more prevalent in Asian cohorts, with variable prognosis [8,11,51]
Surgical Factors	Incomplete/opening of key sinuses and residual disease (missed cells, structures) increase recurrence risk [21,32,33,52,53,54]
	Surgeon’s experience: high-volume surgeons have lower recurrence rates post-ESS [40,55]
	Insufficient postoperative management and inflammation control (poor healing, inadequate steroids) raise recurrence rates [7,22,47,48]
	High postoperative endoscopy scores (e.g., Lund-Mackay) within 3–6 months signal higher likelihood of relapse [7,9,57]

**Table 2 medicina-61-01620-t002:** Predictive Models for Recurrence.

Model	Parameters	Highlights	Limitations
Clinical Predictors and Symptom-Based Models	Comorbidities: Asthma, allergic rhinitis, NSAID-exacerbated respiratory disease (NERD), previous surgery Symptom Trajectory: SNOT-22 change over 3–12 months Surgery Interval: Shorter interval between surgeries	Strong, consistent clinical predictors for recurrence and revision surgery Dynamic symptom tracking (e.g., SNOT-22) offers added prognostic value	Often region and tool- dependent; symptoms alone have moderate accuracy [7,30,58,59,60,61,62,63,64,65,66,67]
Serologic & Immunologic Biomarkers	Peripheral eosinophil, basophil counts, ELR, NLR, serum ECP Cytokines: Interleukins (e.g., IL-5, IL-13), eotaxin, complement proteins Immune cell subsets: Regulatory T cells, innate lymphoid cells	Eosinophil counts (peripheral and tissue) reproducibly linked with recurrence Cytokines/immune markers refine endo- typing and risk assessment	Cut-offs not standardized; moderate predictive accuracy [48,68,69,70,71,72,73,74,75,76,77,78]
Histologic Markers	Tissue eosinophilia (e.g., >55/HPF or ≥27%) Charcot-Leyden crystals IL-5, IL-13 expression, dense eosinophil/mast cell infiltration Eosinophilic mucin	Robust predictor in Asian cohorts Mast cell burden linked to early recurrence	Thresholds debated; invasive sampling often required [9,59,70,79,80,81,82,83,84]
Radiologic Predictors	Lund-Mackay CT score (LM) Ethmoid-to-maxillary opacification ratio Radiologic endotypes (e.g., CCAD, AFRS, AERD forms)	Higher LM/ethmoid dominance: poorer prognosis CCAD: lower recurrence despite ethmoid- predominance	Severity interpretation must consider disease endotype [85,86,87,88,89,90]
Microbiota	Staphylococcus aureus colonization Specific nasal/rectal microbial signatures	*S. aureus* carriage linked to higher recurrence risk Microbial diversity as possible risk modifier	Research emerging; not yet in routine use [91,92,93]
Integrated and Machine Learning (ML)-Based Models	Multivariable models combining: Symptoms (VAS, SNOT-22), radiology (LM), comorbidities, eosinophils, cytokines ML using miRNAs, cytokines, medication history	Superior predictive accuracy (e.g., >80%) Asia: logistic regression, nomograms Western: multivariable and ML models	No single biomarker suffices; integration improves accuracy [7,18,94,95]

**Table 3 medicina-61-01620-t003:** Comparison of CRS Features Between Western and Asian Populations.

Feature	Western Populations	Asian Populations	Clinical Implications
Dominant Inflammatory Endotype	Predominantly type 2 (eosinophilic) inflammation	More heterogeneous; higher non-type 2 or mixed eosinophilic-neutrophilic	Endotype-driven treatment essential; differences affect biologic response and prediction models [5,6,24,46]
Asthma Prevalence	High, strong association with recurrence	High, especially in combination with allergic rhinitis	Asthma co-management improves outcomes; must be factored into risk models [18,24,35,40,46,47,48,50]
AERD Prevalence	Higher prevalence; linked to severe disease and >50% recurrence	Less frequently reported; role still relevant	Predicts severe, treatment-resistant CRS; influences surgery and biologic choice [3,21,47,94]
Recurrence Rates	12–76.6%, often >30% in high-risk groups	13.6–65.3%, most between 25–45%	Regional data must be interpreted in context of definitions and follow-up [11,12,13,14,15,16]
Risk Factors	Asthma, AERD, eosinophilia, CF, female sex, smoking	Tissue/blood eosinophils, uric acid, metabolic syndrome, BMI, younger age	Tailored risk stratification and monitoring required for comorbidities [3,18,20,22,28,37,39,40,41]
Predictive Models	Integrated models: radiologic scores, symptoms, history, biomarkers; ML-based models emerging	Emphasis on inflammatory biomarkers (e.g., eosinophils, IL-6/IL-8); logistic regression and nomograms	Region-specific models needed; integrated data improves prediction [7,8,18,94,95]
Biologic Use	Widely used; early integration with surgery and topical steroids	Limited use; reserved for type 2 patients due to cost/regulation	Access disparity impacts treatment outcomes and equity [5,6,25,101,115]
Surgical Revision Patterns	More frequent; Guide-lines support earlier revision for T2 inflammation	More conservative; lower revision rates and delayed intervention	Careful assessment of candidacy and timing critical; cultural/systemic constraints matter [20,55,105,106,107,115]

## Data Availability

No new data were created or analyzed in this study.

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
