# Peer review of "Recurrence of Chronic Rhinosinusitis with Nasal Polyps After Surgery: Risk Factors, Predictive Models, and Treatment Approaches with a Focus on Western and Asian Differences"

_medicina, 2025, doi:10.3390/medicina61091620_

Round 1
Reviewer 1 Report
Comments and Suggestions for Authors
The review titled “Recurrence of Chronic Rhinosinusitis with Nasal Polyps After Surgery: Risk Factors, Predictive Models, and Treatment Approaches with a Focus on Western and Asian Differences” provides an overview of CRS recurrence rates following surgery. It highlights differences between Western and Asian patients in terms of recurrence, inflammatory markers, treatment approaches, and predictive models. The article is well written, with a clearly defined methodology, including appropriate inclusion and exclusion criteria, and a well-structured discussion.
Minor suggestion
Please consider including a visual representation (e.g., a graph) to illustrate recurrence rates in Western vs. Asian patients.
Author Response
Comment 1 : Please consider including a visual representation (e.g., a graph) to illustrate recurrence rates in Western vs. Asian patients.
Response1: Response: Thank you for your comment. We have now thoroughly revised the manuscript on page 4, line 144 to l50 , and add the figure 2.
Reviewer 2 Report
Comments and Suggestions for Authors
dear Authors
Good to summarize the literature around this topic. However we find some items could still improve the review:
- Introduction: the definition of chronic rhinosinusitis is not fully mentioned. Represent the diagnostic criteria, which are generally consistent across regions and based on guidelines such as EPOS and ICAR.
- Section 2.2: please specify the age range of the adult patients/pediatric studies. Additionally, some inclusion and exclusion criteria are stating the same condition, please avoid overlap.
- Section 3.2: the terms 'most studies' (line 1) and 'the majority' (line 2) are too vague, please clarify.
- Section 5.5 and 5.6: please mention if there is a distinction between Western and Asian countries.
- Section 6.2: please clarify the differences between radical ESS, ESS combined with Draf 3, conventional FESS, traditional ESS.
- Tables: The layout of the tables could be improved for readability. Please use a consistent style for all tables.
- Table 3: This table should be referenced in the results section. Additionally, it lacks references. Please ensure all data sources are cited.
- Abbreviations: Please clarify the abbreviation 'ML'.
- Please add also more reference about rebout surgery
Author Response
Review 2
Good to summarize the literature around this topic. However we find some items could still improve the review:
- Introduction: the definition of chronic rhinosinusitis is not fully mentioned. Represent the diagnostic criteria, which are generally consistent across regions and based on guidelines such as EPOS and ICAR.
Response: Thank you for your comment. We have now thoroughly revised the manuscript on page 1, line 34 to 43.
Chronic rhinosinusitis (CRS) is a prevalent and debilitating inflammatory disease of the paranasal sinuses, characterized by persistent sinonasal symptoms lasting for at least 12 weeks. The diagnostic criteria, which are generally consistent across regions and based on guidelines such as the European Position Paper on Rhinosinusitis and Nasal Polyps (EPOS) and the International Consensus Statement on Allergy and Rhinology: Rhinosinusitis (ICAR), require at least two of the following symptoms: nasal blockage/obstruction/congestion or discharge (anterior/posterior nasal drip), facial pain/pressure, or reduction/loss of smell. These must be supported by objective evidence of inflammation, such as nasal polyps, mucopurulent discharge, or edema on endoscopy, or mucosal changes on computed tomography (CT) within the ostiomeatal complex and/or sinuses.
- Section 2.2: please specify the age range of the adult patients/pediatric studies. Additionally, some inclusion and exclusion criteria are stating the same condition, please avoid overlap.
- Response: Thank you for your comment. We have now thoroughly revised the manuscript on page 2, line 78 to 90.
Inclusion criteria:
- Studies involving adult patients (aged ≥18 years) undergoing endoscopic sinus surgery (ESS) for chronic rhinosinusitis (CRS)
- Reporting recurrence rates, recurrence risk factors, prediction models, or postoperative treatment strategies
- Conducted in or reporting data from Asian or Western populations
- Published in English with full text available
Exclusion criteria:
- Pediatric studies (involving patients <18 years)
- Case reports or case series with <30 patients
- Narrative reviews without systematic methodology, unless cited for background
- Conference abstracts, editorials, and letters
- Section 3.2: the terms 'most studies' (line 1) and 'the majority' (line 2) are too vague, please clarify.
- Response: Thank you for your comment. We have now thoroughly revised the manuscript on page 4, line 122 to 130.
- In Asian cohorts, recurrence rates ranged from 13.6% to 65.3%, with 8 out of 12 studies reporting between 25% and 45%. Of these, 10 out of 12 were retrospective with short- to intermediate-term follow-up (typically 6–36 months). However, two recent studies have expanded this landscape. A multicenter retrospective study which followed up for 5 years across five Asian countries reported recurrence rates of 63.3% in Korea and 33.3% in Japan, underscoring the variability within Asia [11]. Meanwhile, a prospective cohort study with an 8-year follow-up in China showed an overall recurrence rate of 21.8%, but noted a significantly higher rate (72.7%) among patients with type 2 inflammatory endotypes [12].
- Section 5.5 and 5.6: please mention if there is a distinction between Western and Asian countries.
- Response: Thank you for your comment. We have now thoroughly revised the manuscript on page 9, line 345 to 348, and line 358-359
The E/M opacification ratio is a more consistent predictor for recurrence in Asian/non-type 2 CRS and a marker of olfactory recovery in Western/type 2 CRS populations. Radiological severity predicts recurrence in both regions but with different modifying factors.
- aureus and microbiota data suggest region-specific patterns influencing recurrence.
- Section 6.2: please clarify the differences between radical ESS, ESS combined with Draf 3, conventional FESS, traditional ESS.
- Response: Thank you for your comment. We have now thoroughly revised the manuscript on page 11, line 414 to 426.
Endoscopic sinus surgery (ESS) encompasses a range of techniques for treating chronic rhinosinusitis. Radical ESS is characterized by extensive mucosal and bone removal, aiming to eradicate disease and minimize recurrence, and is generally reserved for refractory or severe cases. The Draf 3 procedure, typically performed in conjunction with ESS, involves a modified Lothrop frontal sinusotomy to achieve maximal frontal sinus drainage, especially for complex or recurrent frontal sinus pathology. Conventional functional endoscopic sinus surgery (FESS) and traditional ESS are less extensive, focusing on restoring natural sinus ventilation by selective removal of diseased tissue, which results in faster recovery but higher early recurrence. Reboot surgery is a more radical, non-mucosa-sparing technique involving complete mucosal removal down to the periosteum, targeting refractory chronic rhinosinusitis with nasal polyps (CRSwNP) unresponsive to previous therapies, significantly lowering polyp recurrence and improving long-term outcomes.
- Tables: The layout of the tables could be improved for readability. Please use a consistent style for all tables.
Response: Thank you for your comment. We have now thoroughly revised the tables.
- Table 3: This table should be referenced in the results section. Additionally, it lacks references. Please ensure all data sources are cited.
- Response: Thank you for your comment. We have now thoroughly revised the table 3. Page 13.
- Abbreviations: Please clarify the abbreviation 'ML'.
Response: Thank you for your comment. We have now thoroughly revised the table 15. Abbreviation
- Please add also more reference about rebout surgery
Response: Thank you for your comment. We have now thoroughly revised the concept in Page 13 Line 423-426.
